# The Effect of Ne⁺ Ion Implantation on the Crystal, Magnetic, and Domain Structures of Yttrium Iron Garnet Films

Igor Fodchuk [1], Andrij Kotsyubynsky [2], Andrii Velychkovych [3,*], Ivan Hutsuliak [1], Volodymyra Boychuk [4], Volodymyr Kotsyubynsky [2] and Liubomyr Ropyak [5]

[1] Department of Solid State Physics, Yuriy Fedkovych Chernivtsi National University, 2 Kotsiubynsky Str., 58012 Chernivtsi, Ukraine
[2] Department of Material Science, Vasyl StefanykPrecarpathian National University, 57 Shevchenko Str., 76018 Ivano-Frankivsk, Ukraine
[3] Department of Construction and Civil Engineering, Ivano-Frankivsk National Technical University of Oil and Gas, 15 Karpatska Str., 76019 Ivano-Frankivsk, Ukraine
[4] Department of Physics, Vasyl StefanykPrecarpathian National University, 57 Shevchenko Str., 76018 Ivano-Frankivsk, Ukraine
[5] Department of Computerized Engineering, Ivano-Frankivsk National Technical University of Oil and Gas, 15 Karpatska St., 76019 Ivano-Frankivsk, Ukraine
* Correspondence: a_velychkovych@ukr.net

**Abstract:** The mechanism of the influence of crystal inhomogeneities on the magnetic and domain microstructures of functional materials based on yttrium iron garnet heterostructures is an important subject of investigation due to the aim to predict parameters for manufacturingpurposes. A study of the structural and magnetic characteristics of a set of yttrium iron garnet films grown on gadolinium–gallium garnet substrate is presented. High-resolution X-ray diffractometry, Mössbauer spectroscopy, MFM, as well as ion implantation simulation and X-ray diffraction simulation were used together to determine the features of the effect of Ne⁺ ion implantation with different dose rates on the samples. The simulation of ion implantation with $E = 82$ keV showed energy loss profiles of Ne ions with subsequent defect formation up to amorphization of near-surface layers at high doses. Implantation creates two magnetically non-equivalent types of tetrahedrally located $Fe^{3+}$ ions, which leads to a rotation of the total magnetic moment relative to the film surface and a change in the width of the magnetic domain stripes.

**Keywords:** yttrium iron garnet; epitaxial films; magnetic structure; X-ray diffraction; Mössbauer spectroscopy; ion implantation





## 1. Introduction

Innovative technologies for the use of single-crystal materials open up new opportunities for creating highly sensitive magnetic sensors [1,2], for assessing mechanical deformations and stresses in thin-walled structural elements [3–5], optical sensors for monitoring the opening of cracks and quality control of precision surface treatment [6–9] and the production of special films [10].

The search for new materials with specified physicochemical properties [11] is one of the urgent problems of modern optoelectronics and the chemistry of optical materials. Garnets are well-known as scintillation and phosphor materials and when doped with other transition elements can have many other important applications such as materials for solid-state lasers, thermoluminescent dosimeters, thermometers, etc.

The work [12] considers the deformation control of magnetic anisotropy in iron-garnet-yttrium films in a composite structure with a substrate of alumino-yttrium-garnet. The paper [13] summarizes the latest results of the development of composite luminescent materials based on single crystal films and single crystals of simple and mixed garnet compounds obtained by growing by liquid phase epitaxy. Crystallization and study of

the structural and optical properties of monocrystalline film phosphors doped with rare earth elements are considered in [14]. In [15], a new process was developed for depositing magnetic garnet on a flexible substrate by applying already crystallized magnetic garnet powders. The results are used to produce flexible magneto-optical indicators. In [16], a periodic band-like structure of nanoparticles was created on the surface of a thin single-crystal film of yttrium-iron garnet grown on a gallium-gadolinium garnet substrate by the immersion method. It was shown that such a periodic structure causes the formation of forbidden bands in the transmission spectra of magnetostatic surface spin waves, which cannot be achieved by other types of filters [17].

To incorporate yttrium garnet into nanofabrication processes, it is necessary to fabricate very thin films while maintaining the exceptional magnetic properties of the material. In [18], various results published over the past decade in this area are considered and discussed. In [19], a review of electron detectors with garnet scintillators was carried out and attention was drawn to the prospects and limitations of their practical use.

An analysis of the magnetic ordering character in iron-rich phases [20] with ionicor covalent bonds, which can be used as reinforcements in metal matrix composites or composite coatings [21], also plays an important role in terms of their elastic constants and thermodynamic stability [22–24].

Functional materials based on single-crystal garnets are currently used in spintronics and magneto-optic devices [25]. Garnet films for magnon spintronics and magnon-optical hybrid systems are of particular interest today, and the effect of crystal properties on its magnetic and domain structures becomes critically important for obtaining materials with controllable predictable characteristics [26]. The presence of interconnected magnetic and elastic sublattices makes it possible to obtain a high-quality oscillating system. The control of the structural characteristics of yttrium iron garnet (YIG) epitaxial films grown on a gadolinium-gallium garnet (GGG) substrate becomes of key importance. Ion implantation is used for local modification of the structural and magnetic properties of YIG [27]. The formation of a structurally disordered near-surface layer with a controllable defect distribution profile makes it possible to lower the threshold for the excitation of spin waves and initiate nonlinear self-phase modulation [28]. Ion implantation makes it possible to reduce the length of exchanged spin waves and optimize magnetic losses, which is important for the high performance of spintronic devices [29].

However, there are no detailed experimental studies of the effect of crystal inhomogeneities on the magnetic and domain microstructures of near-surface YIG layers. These circumstances became the motivation for our studies.

The aim of this paper is to experimentally study the effect of crystal inhomogeneities on the magnetic and domain microstructures of the near-surface layers of yttrium iron garnet films grown on a gadolinium-gallium garnet substrate.

## 2. Materials and Methods

Single-crystal $Y_3Fe_5O_{12}$ (YIG) films 2.85 μm thick were grown on $Gd_3Ga_5O_{12}$ GGG substrates by liquid phase epitaxy (growth temperature was 1248 K, overcooling temperature was 14 K). The (111)-oriented GGG single crystal substrates were used. YIG films were implanted with $Ne^+$ ions ($E$ = 82 keV) with doses of 0 (sample D0), $1 \times 10^{14}$ cm$^{-2}$ (sample D1), $2 \times 10^{14}$ cm$^{-2}$ (sample D2) and $4 \times 10^{14}$ cm$^{-2}$ (sample D3). Implantation current less than $2 \times 10^{-5}$ did not allow self-annealing of radiation defects.

Defects of the crystal structure of YIG crystals were studied by X-ray diffraction methods on anX'Pert PRO MRD XL X-ray diffractometer with CuKα1 radiation. The rocking curves for the (444) and (888) crystal planes were recorded using the $\omega/2\theta$ scan mode for triple-crystal geometry and the $\omega$-scan mode for double- and triple-crystal sets [30].

Mössbauer spectroscopy was used to study magnetic and electrical superfine interactions in implanted YIG films. A $^{57}$Co source in a chromium matrix with an activity of about 70 mKu was used. The spectra were calibrated relative to α-Fe. Zero-level instability and

registration error did not exceed 0.05 mm/s, the line width for α-Fe was 0.29 mm/s. The exit depth of conversion electrons did not exceed 90 nm, and about 2/3 of all registered signals were obtained from a near-surface layer with a thickness of <65 nm.

The magnetic domain structures were observed by magnetic force microscopy(MFM) using a NanoScope IIIa Dimension 3000 (Veeco Instruments Inc., Plainview, NY, USA) scanning probe microscope [31].

## 3. Results and Discussion

### 3.1. Simulation of Defect Formation during Ion Implantation of YIG Crystals with Ne⁺ Ions

The slowdown of a high-energy ion in a crystal is caused both by elastic collisions with atomic nuclei and inelastic interaction with electronic shells. The mechanism of defect generation in elastic collisions is well known, and the simulation of energy loss by ions during target damage and ionization was performed using the SRIM-2013 software package [32]. Experimental conditions with an angle between the direction of the ion beam and the normal vector to the surface of 12° were used as a precondition for simulation, which makes it possible to prevent channeling effects. The used threshold displacement energies ($E_d$) of $Y^{3+}$, $Fe^{3+}$ and $O^{2-}$ ions from regular positions in the YIG lattice are 66, 56 and 40 eV, respectively [33]. The simulation results show that the stopping projective range ($R_p$) of Ne⁺ ions ($E$ = 82 keV) is 106 nm with a standard deviation $\Delta R_p$ = 43 nm (Figure 1a).

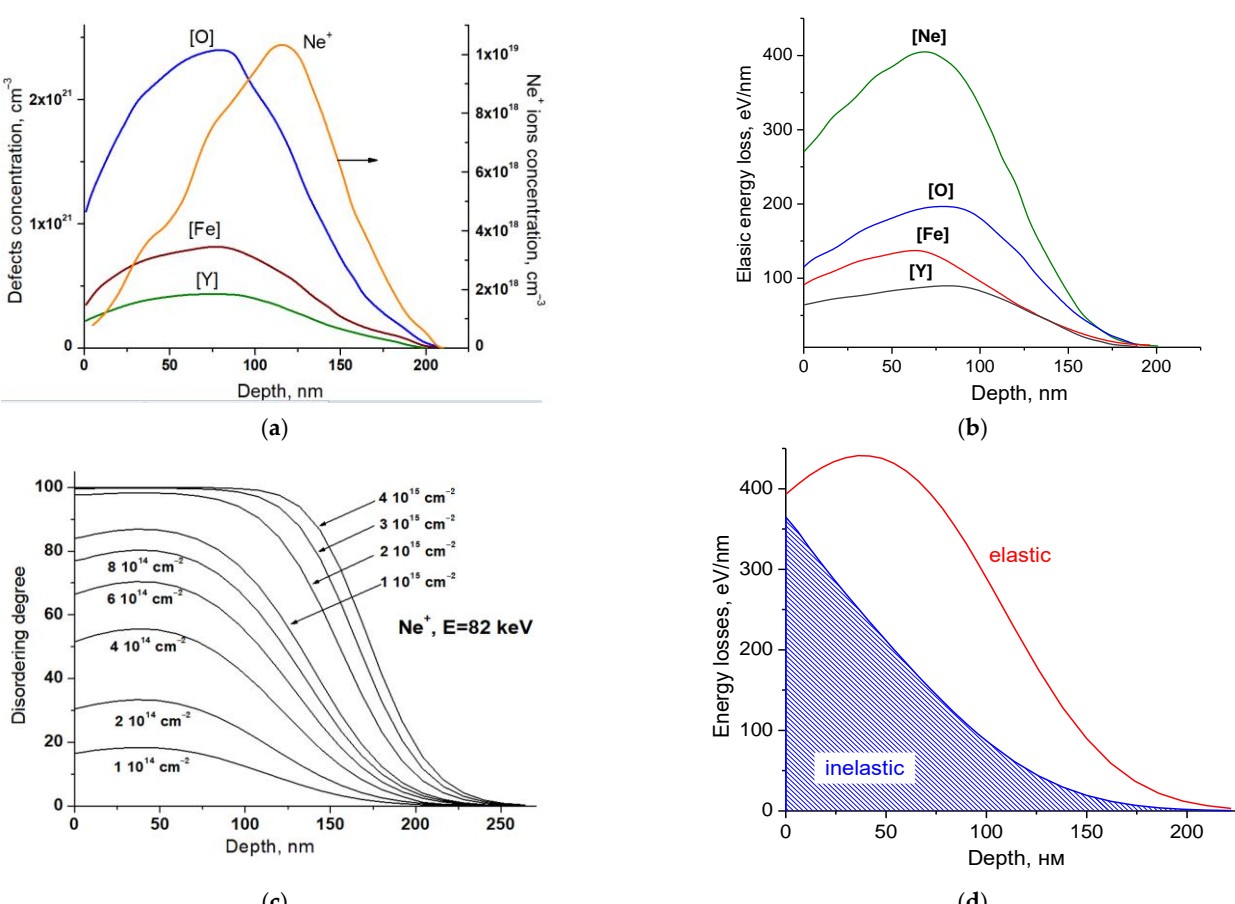

**Figure 1.** Simulation of implantation of Ne ions ($E$ = 82 keV) in YIG: Ne⁺ ion distribution and defect concentration profiles (**a**); energy loss profiles (**b**); comparison of elastic and inelastic energy losses for Ne⁺ ions (**c**); evolution of the disordering degree during implantation with increasing dose (**c**) (the SRIM (**a**,**b**) and SUSPRE (**c**,**d**) software were used).

The average track length of implanted ions is about 210–230 nm. The concentration of implanted ions at an irradiation dose $D = 1 \times 10^{14}$ cm$^{-2}$ is about $1 \times 10^{19}$ cm$^{-3}$ (ion concentration in YIG is $8.4 \times 10^{22}$ cm$^{-3}$). The concentration of primary-displaced atoms for the irradiation dose $D = 1 \times 10^{14}$ cm$^{-2}$ is about $2.3 \times 10^{21}$ cm$^{-3}$. The highest efficiency of defect formation is observed at a depth of $75 \pm 15$ nm, where the probability of overlapping ion tracks and the formation of defects is maximum. It is in this region that the maximum of the elastic energy distribution occurs, which for Ne$^+$ ions is about 420 eV/nm (Figure 1b). According to the Kinchin–Pease model, the formation of radiation defects is possible when the energy transferred to the target atom during elastic interaction exceeds $2E_d$. As a result, defect formation in the anion sublattice ($E_d = 40$ eV) causes the appearance of about 4–5 radiation defects per longitudinal path corresponding to the lattice constant of the garnet crystal. Secondary defects are generated by displaced target atoms at a depth of about 200 nm. According to the statistical analysis of the simulation results, the deceleration of Ne$^+$ ions leads to the formation of tracks of displaced target atoms. The generation of a Frenkel pair as a primary defect is the most probable (about 44%), while the probability of developing a cascade with 2 recoil atoms is 13%, with 3 atoms—6%, with 4 atoms—4%, etc.

The overlap of separate cascades is most likely in the depth range of 65–85 nm, where the elastic energy losses are the greatest (the average volume of disordered clusters is 0.15 nm$^3$). Amorphization of the structure in this region will begin with an increase in the implantation dose, followed by expansion of its area both towards the surface and deep into the film [34]. The projected range algorithm was calculated as a solution of the Boltzmann transport equation, and the SUSPRE software [35] was used to simulate the structural disorder of YIG after Ne$^+$ implantation with ion doses in the range of $1 \times 10^{14}$–$4 \times 10^{16}$ cm$^{-2}$ (Figure 1c). At the same time, the disordering of the YIG crystal structure during ion implantation can also start from the film surface [30], where ion sputtering processes are observed and the inelastic energy losses of the implant are the greatest. An additional factor for it is the migration of radiation defects to the film surface. The average value of the inelastic energy loss of the Ne$^+$ ion in the YIG structure is about 250–270 eV/nm, and that is about 46% of elastic losses (Figure 1d). The energy transferred to the electron subsystem has a maximum at the initial stage of the projective range (of about 360 eV/nm). Relaxation of an excited electron subsystem in ionic crystals with the generation of point defects can occur by two typical mechanisms: electrostatic [36] and vibrational [37]. A precondition for the implementation of the first mechanism is the excess energy of the electrostatic interaction of short-lived charged particles generated during fast ion stopping. The second mechanism is associated with the excitation of valence electrons of the O2p and Fe3d states of target ions with their transition to the conduction band (the band gap of YIG is 3.1 eV [38]). The stopping ion creates an excitation channel around the track with a diameter equal to the electron diffusion length during the relaxation time $\tau$. The value of $\tau$ can be estimated as $\tau = r^2/\chi$, where $r$ is the ion track radius and $\chi = 5.8 \times 10^{-11}$ m$^2$/s is thermal conductivity of YIG. Thermodynamically stable defects can appear if $r$ exceeds the average distance between ions in the YIG lattice (about 0.18 nm). As the result, the duration of such a process should exceed $7 \times 10^{-10}$ s, but the oscillation time of an atom in a crystal lattice site is about $10^{-13}$ s. In this case, the oscillatory mechanism of defect formation in the YIG structure at applied energies of the Ne$^+$ ion implantation is unlikely compared with the electrostatic mechanism. Relaxation of the electron subsystem after impact ionization of the internal electron shells of a crystal lattice ion can cause the formation of point defects if the energy of the Coulomb interaction of a short-lived electron excitation with the charge of the lattice ions exceeds the binding energy [34]. The process precondition is satisfied if the lifetime of an ionized atom ($\tau_i$) exceeds the characteristic time ($\tau_{dis}$) of formation of an interstitial atom (when lattice atom is transferred to an interstitial position), which is determined by the lattice vibration period. For wide bandgap dielectric materials with ionic-covalent chemical bonds of the YIG type, the condition $\tau_{dis} \ll \tau_i$ is fulfilled [39]. The energy criterion for defect formation is satisfied by multiple ionization of the oxygen anion of the crystal lattice due to Coulomb repulsion with a close surrounding. The ionization

energy of the K shell of $O^{2-}$ is about 800 eV, and since the maximum inelastic energy loss in the investigated case is about 360 eV/nm, it can be assumed that ionization of both the K and L shells is possible (and this is more likely for L-electrons). The differential cross sections for the inelastic ionization of the L shell can be calculated using the approach [40].

### 3.2. High-Resolution X-ray Diffractometry

An analysis of the intensity distribution of scattered X-rays as a function of reciprocal space coordinates (reciprocal space map, RSM) using the kinematic theory of scattering in real crystals [41] makes it possible to observe the evolution of the stress state in a YIG film after various implantation doses. RSMs are plotted with coordinates ($q_y$, $q_h$), where the $q_y$ direction is normal to the film surface, and $q_h$ coincides with the direction of the reciprocal lattice vector for a certain reflection (Figure 2).

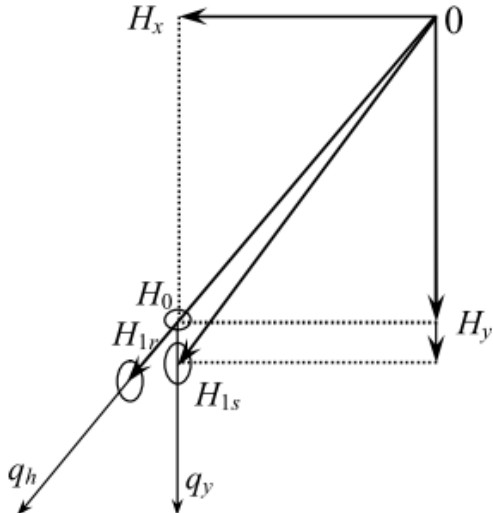

**Figure 2.** The reciprocal lattice sites of the GGG substrate ($H_0$) and the YIG film in the relaxed($H_{1r}$) and unrelaxed ($H_{1s}$) states.

The intensity distribution of X-ray scattering around (444), (888) and (880) nodes obtained for the initial (D0) and implanted (D1, D2, D3) samples wereanalyzed using the common kinematic theory for defective crystals (Table 1).

The relatively large thickness of the YIG film (2.85 µm) caused a significant decrease in the peak intensity of the substrate in the case of symmetrical reflection. The intensity of coherently scattered X-rays for a non-implanted sample in the direction normal to the film surface (distributed along the $q_y$ axis) is typical of a heterostructure in a state close to unrelaxed. Asymmetric reflection (880) was used to determine the relaxation degree as a shift of the intensity maxima of scattered X-rays between $q_y$ and $q_h$ directions. The RSMs obtained for a non-implanted film demonstrate the presence of mechanical stress due to pseudomorphic film growth with a possible change in symmetry from cubic to rhombohedral by the appearance of a third-order symmetry axis perpendicular to the plane of the heterostructure.

The weak broadening of the diffraction pattern in the region of the peak toward $q_h$ for the non-implanted sample is most likely due to the presence of point defects such as oxygen ion vacancies formed in the near-surface layer at the final stages of film growth. Ion implantation with a dose of $1 \times 10^{14}$ cm$^{-2}$ (sample D1) causes a decrease in this broadening, which corresponds to an increase in structural ordering. The observed phenomena can be explained by the partial relaxation of internal strains in the near-surface layer due to the initiation of dislocation migration to the film surface. Increasing the implantation dose to $2 \times 10^{14}$ cm$^{-2}$ (sample D2) causes a broadening of the RSM intensity distribution due to the formation of defect clusters as a result of an increase in the concentration of radiation

defects in the superficial layer with a thickness of about 200 nm. The decreasing intensity broadening on RSM for sample D3 implanted with Ne$^+$ ions at a dose of $4 \times 10^{14}$ cm$^{-2}$ indicates partial amorphization of the disordered layer.

**Table 1.** RSMs for YIG/GGG films implanted with Ne$^+$ ions ($E$ = 82 keV) with doses of 0, $1 \times 10^{14}$, $2 \times 10^{14}$ and $4 \times 10^{14}$ cm$^{-2}$ obtained for (444), (888) and (880) reflexes.

| Sample (Dose, cm$^{-2}$) | Reflexes | | |
|---|---|---|---|
| | **(444)** | **(888)** | **(880)** |
| D0 (0) | | | |
| D1 ($1 \times 10^{14}$) | | | |
| D2 ($2 \times 10^{14}$) | | | |
| D3 ($4 \times 10^{14}$) | | | |

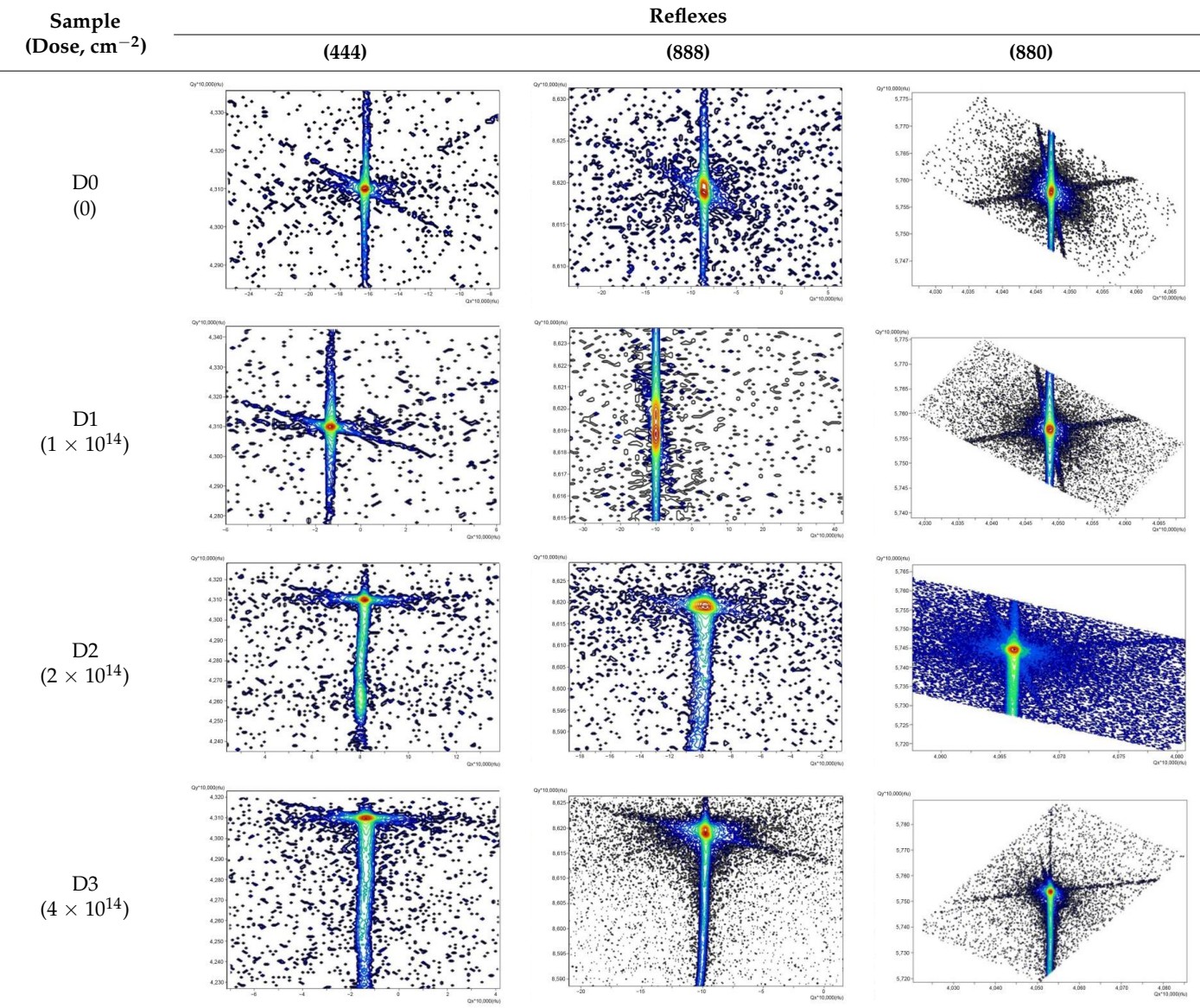

The process of structural rearrangement of the films was studied using X-ray rocking curves obtained for (888) reflection and fitted with two Lorenzian functions (Figure 3a–d).

For sample D1, compared with the non-implanted D0, a sharp increase inthe difference between the angular positions of the reflexes of the YIG film and GGG substrate, which is proportional to the integral value of the mechanical stress in the heterostructures, was observed (Figure 3e), and is the result of relaxation in the near-surface layers. An increase in the implantation dose causes a significant decrease in stress (as for sample D2 with the distribution of defects over implanted layers) with a subsequent increase (for sample D3 with disordered superficial layers). The evolution of the FWHM values of the (888) reflexes of the YIG films also agrees with the results of the RSM analysis (Figure 3f).

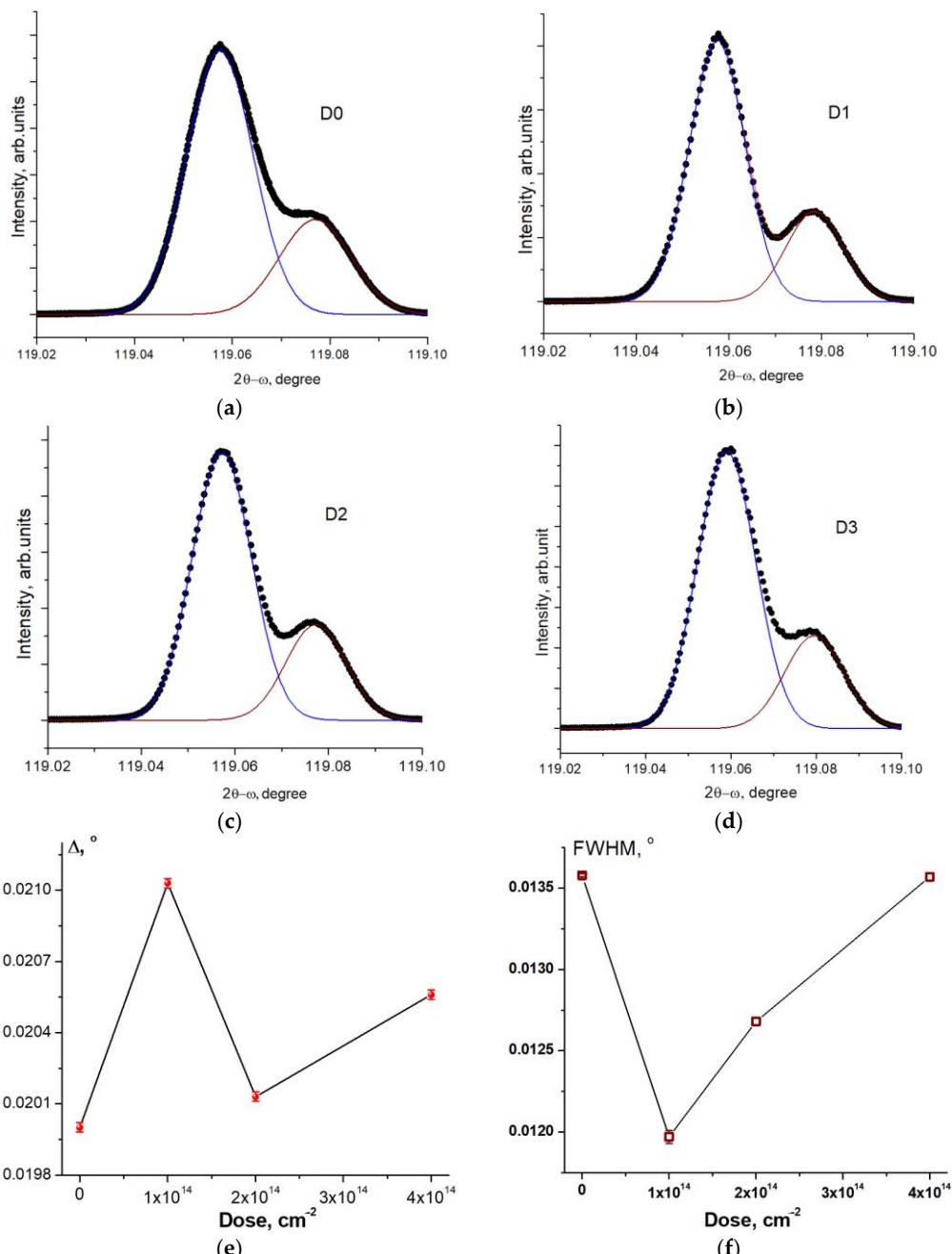

**Figure 3.** Rocking curves ((888) reflection) of YIG/GGG heterostructures implanted with Ne$^+$ ions ($E$ = 82 keV) with doses of 0, $1 \times 10^{14}$, $2 \times 10^{14}$ and $4 \times 10^{14}$ cm$^{-2}$ (dots—experimental data, line—fitting) (**a**–**d**); dependences of the angular differences between the YIG and GGG reflexes (**e**) and the FWHM of the YIG reflex (**f**) on the ion implantation dose.

The intensity oscillations observed on the rocking curves ((444) reflex) of samples D1 and D2 are due to the interference of X-rays with damaged near-surface layers with defect-induced changes in the interplanar distance $\Delta d$. Simulation of rocking curves using the approach [42] makes it possible to obtain the dependence $\Delta d/d_{\mathrm{o}}(z)$ (strain profile), where $d_0$ is the value of the (444) interplanar distance for bulk YIG and $z$ is the distance from the surface. It was assumed that defects are formed both due to elastic nuclear collisions and due to the relaxation of electronic excitations. The strain profile component caused by elastic collisions was described by the asymmetric Gaussian function:

$(\Delta d/d)_n = (\Delta d/d)_n^{(max)} \exp\left[-(z-p)^2/\sigma_n^2\right]$, where $(\Delta d/d)_n^{(max)}$ is the maximum strain observed at $z = p$, $\sigma_n = \begin{cases} \sigma_1, z \le p \\ \sigma_2, z > p \end{cases}$. The strain profile formed due to the relaxation of electronic excitations was described as $(\Delta d/d)_e = (\Delta d/d)_e^{(max)} \exp\left[-(z-q)^2/\sigma_e^2\right]$. Fitting of experimental rocking curves to the theoretical ones made it possible to calculate $(\Delta d/d)_n^{(max)}$, $(\Delta d/d)_e^{(max)}$, $p$, $\sigma_1$, $\sigma_2$ and $\sigma_e$ parameters [43]. The calculation results for samples D1 and D2 are shown in Figure 4.

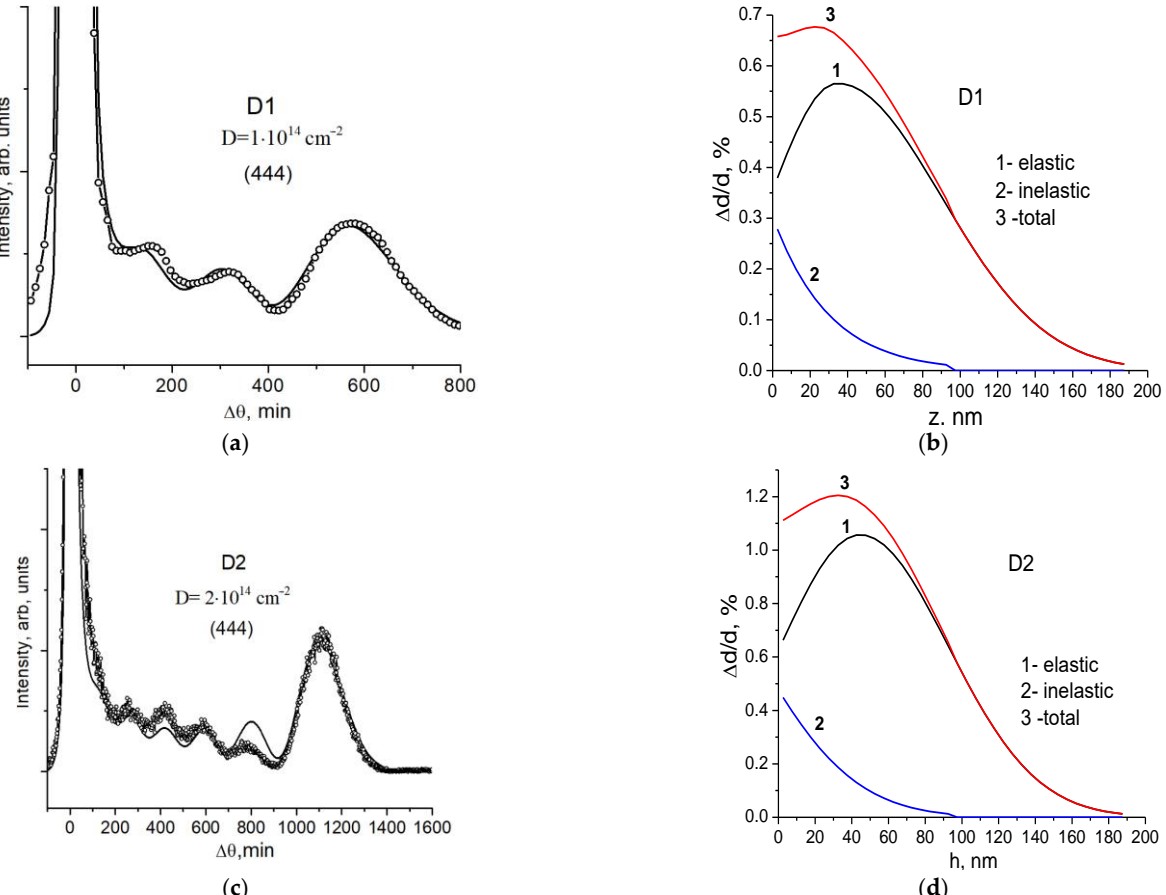

**Figure 4.** Experimental (circles) and calculated (solid lines) rocking curves ((444) reflection) of YIG/GGG heterostructures implanted with $Ne^+$ ions ($E$ = 82 keV) with doses of $1 \times 10^{14}$ cm$^{-2}$ (**a**) and $2 \times 10^{14}$ cm$^{-2}$(**c**), as well as the corresponding strain profiles (**b**) and (**d**), respectively.

The thickness of the damaged layer, determined from X-ray data, correlates with the results of the simulation of the ion implantation process (Figure 1a). The shift in the position of the maximum of the total strain profile relative to other profiles to the surface compared with the maximum concentration of radiation defects is the result of relaxation effects. In sample D3, implanted with a dose of $4 \times 10^{14}$ cm$^{-2}$, no oscillating structure was observed due to amorphization of the near-surface layer of the garnet film.

### 3.3. Mössbauer Spectroscopy and Magnetic Force Microscopy

The crystal structure of YIG (space group Ia3d, $O_h^{10}$) is formed by oxygen coordination polyhedra of three types—24 tetrahedra and 16 octahedra (occupied by $Fe^{3+}$ ions in the *d*- and *a*-positions, respectively) and 24 dodecahedra (occupied by $Y^{3+}$ in c-position). The magnetic order in YIG is formed as a result of the exchange interaction between tetra- and octahedrally located $Fe^{3+}$ ions. The magnetic moments of $Fe^{3+}$ ions in the *a*-and *d*-positions ($M_a$ and $M_d$, respectively) for an infinite perfect crystal are antiparallel. The

total magnetization of YIG films inclines from the [111] direction (normal to the film surface) due to the surface demagnetization effect. The spatial orientation of the magnetic moments $M_a$ and $M_d$ in the YIG structure was investigated by Mössbauer spectroscopy. The nuclei of iron ions, as well as their electron shells, are in an intracrystalline electric field. The intensities of electric quadrupole and magnetic dipole interactions for YIG are approximately equal [44]. The symmetry of the electric field gradient (EFG) tensor is axially symmetric, since the tetrahedral positions have a symmetry axis of the 4th order, and the octahedral ones have the 3rd order. For $Fe^{3+}$ ions in octahedral positions, four different <111> axes are observed with the same values of the EFG moduli. At the same time, there are three <100> axes for $Fe^{3+}$ ions in tetrahedral positions. The difference between the directions of the effective magnetic field on the $Fe^{57}$ nucleus and the EFG is represented by the angle $\theta$. In general, there are seven different $\theta$ values for the YIG structure, which correspond to 7 partial components of the Mössbauer spectra. The (111)-oriented YIG film has three equal possible EFG directions for all tetra-coordinated $Fe^{3+}$(d) ions ($\theta_{1,2,3}^d = 54°44'$) and two different values for octa-coordinated $Fe^{3+}$(a) ions ($\theta_4^a = 0°$ for $1/4$ and $\theta_{5,6,7}^d = 70°32'$ for $3/4$ of $Fe^{3+}$(a) ions). The model of mixed quadrupole and magnetic interaction was used [45]. A coordinate system was chosen with the $z$ axis parallel to $U_{zz}$, where $\beta$ and $\alpha$ are the polar and azimuthal angles between the directions of the effective magnetic field and the $z$ and $x$ axes, respectively. The experimental Mössbauer spectra were fitted with a combination of Lorentzian components as a result of hyperfine interactions of Hamiltonian diagonalization (Figure 5).

**Table 2.** Fitting parameters of Mössbauer spectra for YIG films implanted with $Ne^+$ ($E$ = 82 keV, $\delta$—isomeric shift, $H_{ef}$—effective magnetic field on the $Fe^{57}$ nucleus, $U_{zz}$—axial components of the electric field gradient tensor, $\beta$—the angle between $H_{ef}$ and $U_{zz}$, $\omega$—linewidth, $S$—integral intensity of the spectra component, $\theta$—the angle between $H_{ef}$ and EFG).

| Sample | $I_s$, mm/s | $H_{ef}$, kOe | $V_{zz} \times 10^{21}$, V/m$^2$ | $\beta$,° | $\omega$, mm/s | $S$, % |
|---|---|---|---|---|---|---|
| | | | $a_1$ ($\theta$ = 70.86667) | | | |
| D0 | 0.31 | 486 | 0.423 | 4.33 | 0.36 | 28.8 |
| D1 | 0.33 | 486 | 0.428 | −3.37 | 0.34 | 27.8 |
| D3 | 0.34 | 480 | −0.488 | −17.89 | 0.55 | 22.7 |
| | | | $a_2$ ($\theta$ = 0) | | | |
| D0 | 0.32 | 473 | 1.974 | 90.00 | 0.36 | 10.3 |
| D1 | 0.34 | 471 | 4.746 | 62.62 | 0.34 | 9.8 |
| D3 | 0.35 | 453 | 34.408 | 59.84 | 0.55 | 7.6 |
| | | | $d_1$ ($\theta$ = 54.7333) | | | |
| D0 | 0.10 | 396 | 4.914 | −49.88 | 0.43 | 29.7 |
| D1 | 0.13 | 397 | 5.265 | −47.43 | 0.38 | 26.8 |
| D3 | 0.16 | 389 | 4.291 | −45.99 | 0.80 | 33.9 |
| | | | $d_2$ ($\theta$ = 54.7333) | | | |
| D0 | 0.32 | 383 | −5.244 | −52.02 | 0.468 | 29.4 |
| D1 | 0.37 | 383 | −3.935 | −48.84 | 0.418 | 33.5 |
| D3 | 0.39 | 381 | −7.963 | −42.99 | 0.842 | 27.7 |
| | | | doublet | | | |
| D0 | 0.285 | – | 2.29 | – | 0.359 | 1.8 |
| D1 | 0.234 | – | 2.28 | – | 0.344 | 2.1 |
| D2 | 0.247 | – | 2.10 | – | 0.554 | 8.1 |
| D3 | 0.24 | – | 2.08 | – | 0.62 | 76 |

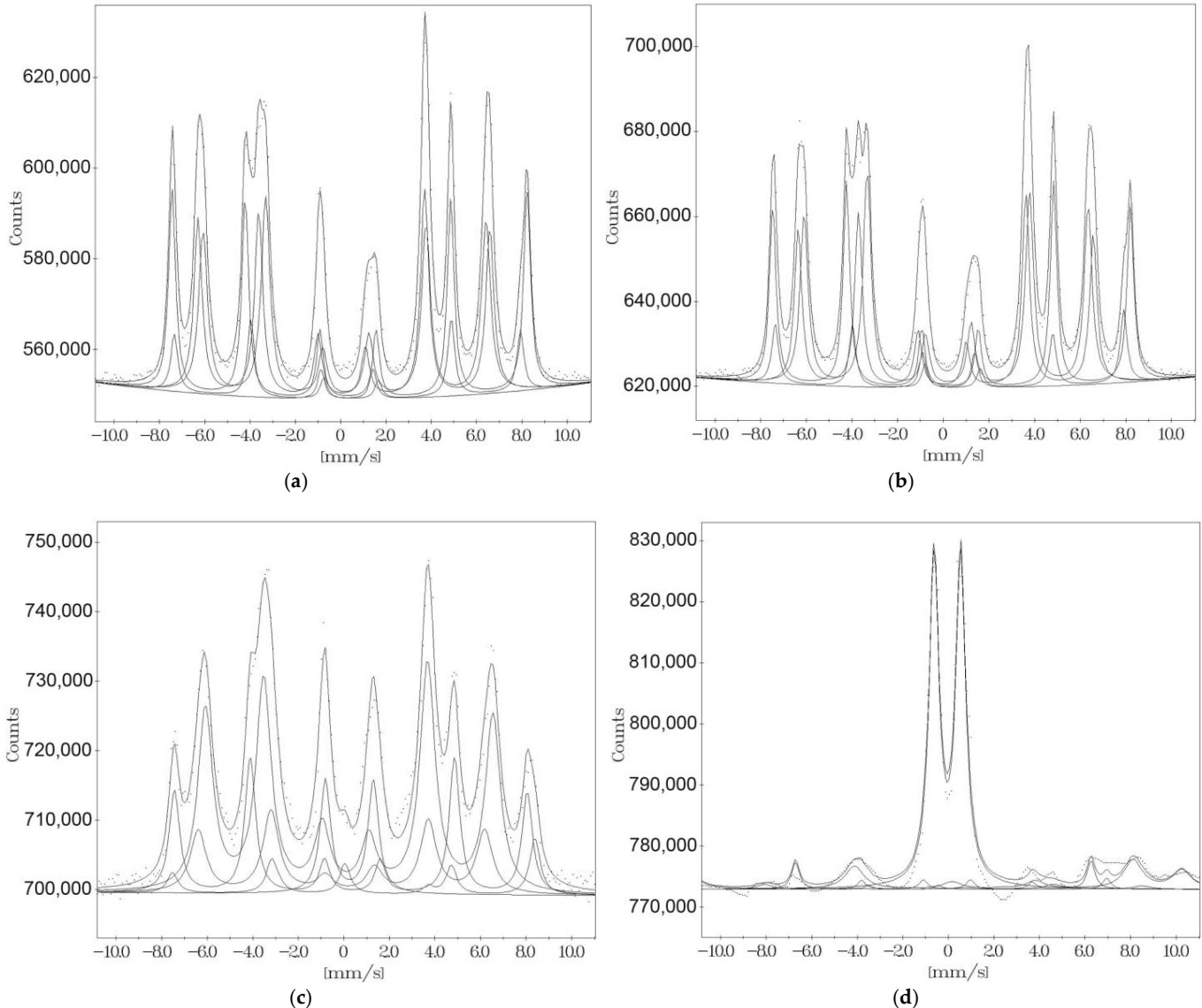

**Figure 5.** Conversion electron Mössbauer spectra (dots- experimental, solid lines- a1, a2, d1, d2 and doublet components) of $Ne^+$-implanted YIG film ($E$ = 82 keV) with doses of 0 $cm^{-2}$, $1 \times 10^{14}$ $cm^{-2}$, $2 \times 10^{14}$ $cm^{-2}$ and $4 \times 10^{14}$ $cm^{-2}$ (**a–d**, respectively). The amplitude, isomer shift δ, magnetic field $H_{ef}$, the value of the axial components $U_{zz}$ of the EFG, and polar angle β between $H_{ef}$ and $U_{zz}$ were used as fitting parameters (Table 2). The calculation was carried out for samples D0, D1 and D2. The Mössbauer spectra of sample D3 mainly consist of a paramagnetic component (about 85% of the integral intensity) due to implantation, which caused crystal disordering, therefore, the calculation of hyperfine parameters for a separate magnetic sublattice is impossible.

The best-fitting results were obtained in the presence of two types of $Fe^{3+}$(d) atoms with different assumed values of the angle β. These distortions can be determined by the presence of anionic defects. Another reason is the uncontrolled entry of impurity atoms into the garnet structure from the melt at the final stages of epitaxial growth. These can be $Pb^{2+}$ and $Pb^{4+}$ ions occupying octahedral positions replacing $Fe^{3+}$, or $Pt^{4+}$ ions occupying only the a-position [46]. Described defects do not have a significant effect on the crystalline order, but a slight distortion of the local environment leads to a change in the direction of the EFG and the absolute values of its axial component, as well as to distortions of the superexchange interaction. All these factors lead to the formation of two non-equivalent $Fe^{3+}$(d) sublattices with non-collinear magnetic moments.

A relatively larger number of oxygen ions in close proximity to $Fe^{3+}$(a) ions increases the probability of Fe(a)-O bond destruction, which manifests itself in a certain increase in the relative content of $Fe^{3+}$(d). The relative occupation of the d- and a-sublattices $n_d/n_a$ for the non-implanted film is 1.51 (the theoretically expected value is 1.50), but an increase in the implantation dose causes an increase in this parameter (Table 2). The decrease in the effective magnetic fields ($H_{ef}$) at the nuclei of $Fe^{3+}$ ions in the a- and d-sublattices with an increase in the implantation dose is observed due to the accumulation of radiation defects and deformation of the Fe–O exchange bonds. A simultaneous increase in the isomeric shift for all sublattices indicates an increase in the Fe–O interionic distance, which leads to an increase in the density of s-electrons at the nuclei of $Fe^{3+}$ ions. A direct relationship is predicted between the relative contribution of 4s electrons to the electronic configuration of the iron ion ($3d4s^x$ configuration) and the values of the isomeric shift δ.

The effective magnetic fields at the nuclei of $Fe^{3+}$ ions and its magnetic moment are antiparallel, therefore, the spatial orientation of the magnetic moments of individual sublattices was calculated. The obtained data correlate with the results of magnetic force microscopy. An increase in the implantation dose leads to a rotation of the total magnetic moment with respect to the film surface (Figure 6) due to magnetostrictive changes in the uniaxial magnetic anisotropy. Mechanical stress is proportional to the dose of radiation defects under implantation conditions without the formation of amorphous clusters (samples D1 and D2). Thus, the obtained results make it possible to predict the magnetic moment orientation as a function of implantation dose. The period of the band domain structure is the smallest for sample D1 with the mostly non-distorted crystal structure (Figure 6). There is a tendency foran increase in the period of band domains with a further increase in the dose, which is associated with an increase in magnetic disordering.

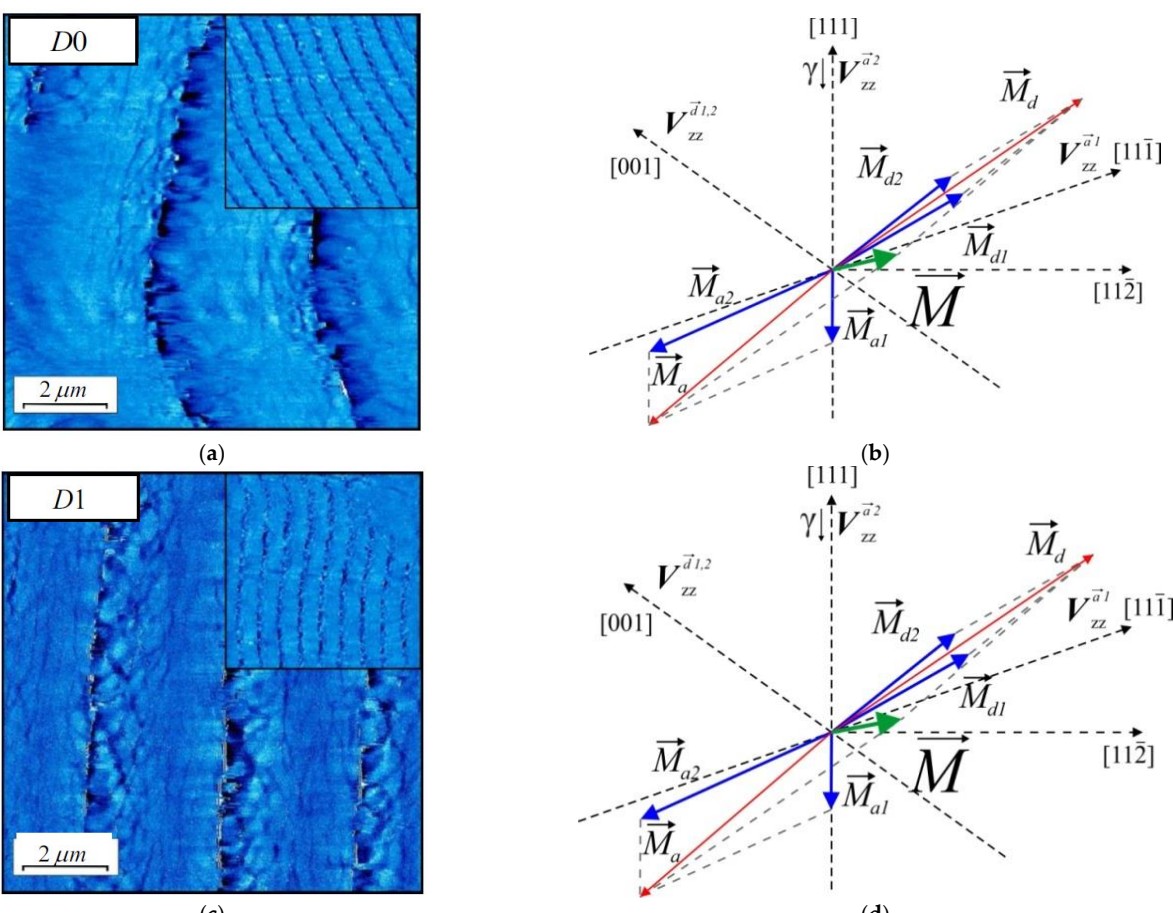

**Figure 6.** *Cont.*

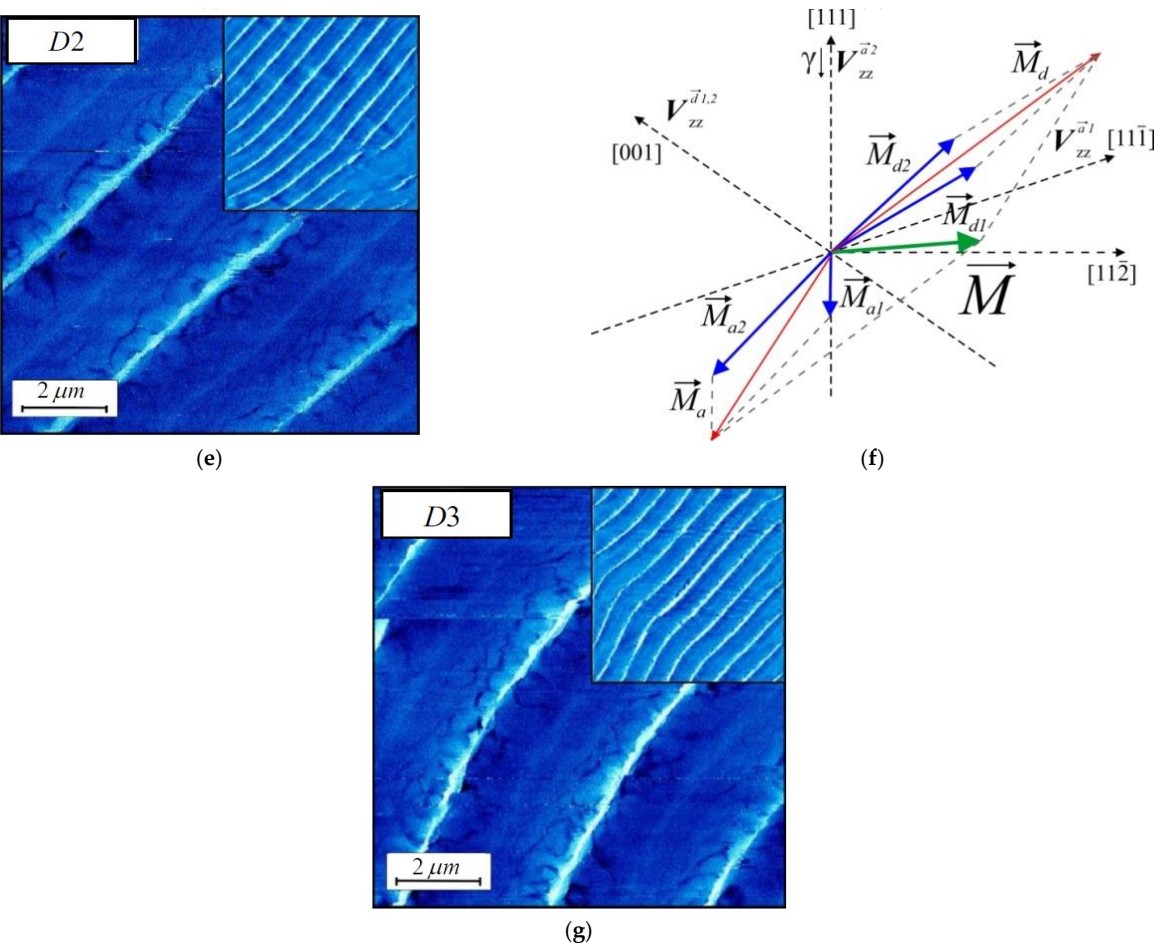

**Figure 6.** MFM images of the band domain structure (**a**,**c**,**e**,**g**) and the corresponding diagrams of spatial orientations of magnetic moments (**b**,**d**,**f**) of Ne$^+$-implanted YIG film (*E* = 82 keV) with doses of 0 cm$^{-2}$, 1 × 10$^{14}$ cm$^{-2}$, 2 × 10$^{14}$ cm$^{-2}$ and 4 × 10$^{14}$ cm$^{-2}$ (samples D0, D1, D2 and D3, respectively).

## 4. Conclusions

Implantation with Ne$^+$ ions (*E* = 82 keV) of monocrystalline epitaxial YIG films grown on a GGG substrate causes the close elastic and inelastic energy losses of implants with the most probable (about 54%) generation of anionic vacancies. The maximum concentration of defect is observed at a depth of 75 ± 15 nm, where the ion tracks overlap, leading to the formation of a structurally disordered area with its further expansion as the implantation dose increases both into depth and into the film surface. The structural ordering of the Ne$^+$-implanted YIG at a dose of 1 × 10$^{14}$ cm$^{-2}$ is correspondingly higher compared with the non-implanted sample (which corresponds to the data of high-resolution X-ray diffractometry and MFM) due to defect migration and mechanical stress relaxation. An increase in the implantation dose leads to an increase in structural disordering, which causes several effects: a decrease in the effective magnetic fields of the nuclei of Fe$^{3+}$ ions, an increase in isomeric shifts, indicating a decrease in the Fe–O chemical bond covalence degree, and rotation of the resulting magnetic moment relative to the film surface. The presence of two magnetically non-equivalent types of tetrahedrally located Fe$^{3+}$ ions was observed for the initial and implanted YIG films as a result of structure distortion. Amorphization of near-surface layers of YIG films was observed at Ne$^+$ implantation dose of 4 × 10$^{14}$ cm$^{-2}$.

**Author Contributions:** Conceptualization, I.F., A.K. and V.K.; methodology, I.F., V.K. and L.R.; software, A.K., A.V., I.H. and V.K.; validation, I.F., A.K., A.V., I.H. and L.R.; formal analysis, I.H., V.B. and V.K.; investigation, I.F., A.K., I.H. and V.K.; resources, I.F., A.K., I.H., V.B. and V.K.; data curation, A.K., I.H., V.B., V.K. and L.R.; writing—original draft preparation, I.F., A.K., A.V., I.H., V.B. and V.K.;

writing—review and editing, I.F., I.H., V.B., V.K. and L.R.; visualization, A.K., A.V., V.B. and V.K.; supervision, I.F. and V.K.; project administration, I.F., V.K. and L.R.; funding acquisition, I.F., I.H., V.B., and V.K. All authors have read and agreed to the published version of the manuscript.

**Funding:** This research was partially supported by the research project of the Ministry of Education and Science of Ukraine 0122U000932 "Mechanisms of structural relaxation and defect formation in heterosystems, thin films and nanocomposite materials".

**Institutional Review Board Statement:** Not applicable.

**Informed Consent Statement:** Not applicable.

**Acknowledgments:** The authors are grateful to the: Ministry of Science and Education of Ukraine for the grant to implement projects 0121U109591 and 0122U002082. The authors are also grateful for the thorough consultations of Vasyl Vytvytskyi Department of Engineering and Computer Graphics, Ivano-Frankivsk National Technical University of Oil and Gas. The team of authors express their gratitude to the reviewers for valuable recommendations that have been taken into account to improve significantly the quality of this paper.

**Conflicts of Interest:** The authors declare no conflict of interest.

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
