# Peer review of "The Effect of Ne+ Ion Implantation on the Crystal, Magnetic, and Domain Structures of Yttrium Iron Garnet Films"

_crystals, doi:10.3390/cryst12101485_

Round 1

Reviewer 1 Report

The authors reported the effect of Ne ions implantation on the crystal, magnetic, and domain structures of yttrium ion garnet films. This work was carefully performed and well discussed. The conclusion was supported by solid data.

High resolution electron microscope images of samples before and after Ne+ implantation should be compared and discussed.

The language quality needs to be improved. There were many grammar mistakes in the main text. For example, 

It was shown that such a periodic structure causes

This article aims to experimental studies

Only a few examples were above listed. The authors need to check the whole draft.

Author Response

All authors would like to thank reviewer for careful revision and evaluation of our paper. We appreciate his/her time and work he/she done. After attentive reading of the comments we complement our article according to reviewer’s remarks.

Reviewer 1

Comments and Suggestions for Authors

The authors reported the effect of Ne ions implantation on the crystal, magnetic, and domain structures of yttrium ion garnet films. This work was carefully performed and well discussed. The conclusion was supported by solid data.

  1. High resolution electron microscope images of samples before and after Ne+ implantation should be compared and discussed.

We agree that high resolution electron microscope studies can in most cases give direct information about structural inhomogeneities in single crystal materials. At the same time, we would like to draw your attention the article is devoted to the study of the ion implantation induced structural changes in the of relatively thick (2.85 mm ) monocrystal Y3Fe5O12  films  grown on thick (of about 500 mm ) Gd3Ga5O12 substrate so HR TEM is not-applicable. On the other hand the HR SEM provides information only about the surface, while we are interested in the defects distribution along implanted layer and their effect on the magnetic microstructure of the film. Accordingly, research using the the magnetic force microscopy is the most optimal

  1. The language quality needs to be improved. There were many grammar mistakes in the main text. For example,

 It was shown that such a periodic structure causes

This article aims to experimental studies

Only a few examples were above listed. The authors need to check the whole draft.

 We took into account all reviewer`s recommendations and consequently corrected all mistakes and incomprehensibilities.  Thank you!

11 October 2022                                                                               Date of article correction

Reviewer 2 Report

In terms of the contents, this work could be published after minor revisions as followed:

There are some typos and grammatical errors, the authors must make an overall revision to improve the readability. For example, “…for   assessing  mechanical 33deformations” should be “…assessing  mechanical 33deformations”; “To introduce Yttrium Garnet into nanofabrication processes it” should be “To introduce Yttrium Garnet into nanofabrication processes is…”

Author Response

All authors would like to thank reviewer for careful revision and evaluation of our paper. We appreciate his/her time and work he/she done. After attentive reading of the comments we complement our article according to reviewer’s remarks.

Reviewer 2

Comments and Suggestions for Authors

In terms of the contents, this work could be published after minor revisions as followed:

There are some typos and grammatical errors, the authors must make an overall revision to improve the readability. For example,

 “…for  assessing  mechanical 33deformations” should be “…assessing  mechanical 33deformations”; “To introduce Yttrium Garnet into nanofabrication processes it…” should be “To introduce Yttrium Garnet into nanofabrication processes is…”

We took into account all reviewer`s recommendations and consequently corrected all mistakes and incomprehensibilities.  Thank you!

11 October 2022                                                                              Date of article correction

Reviewer 3 Report

I recommend tom accepting the manuscript as it is.

Author Response

All authors would like to thank reviewer for careful revision and evaluation of our paper. We appreciate his/her time and work he/she done.

Thank you!

11 October 2022                                                                               Date of article correction

Reviewer 4 Report

While the article itself is fine, a total lack of theoretical investigations of the structure using ab initio methods is slightly disappointing. It may be rather helpful in studying the structure and defects, preventing the manuscript from providing a more significant impact on the field.

Neon ion implantation of YIG is nothing ground-breaking new in the field, and therefore a more precise investigation of what happens exactly would be nice, but maybe out of the scope this time.

While the change in magnetic properties is interesting, it provides minor new insights (but they are there).

Some minor comments to the article:

-) Mössbauer is misspelled
-) "Described defects doesn’t significantly", change to don't
-) the iron ion configuration (3d4sx) is given rather imprecise

Author Response

All authors would like to thank reviewer for careful revision and evaluation of our paper. We appreciate his/her time and work he/she done. After attentive reading of the comments we complement our article according to reviewer’s remarks.

Reviewer 4

Comments and Suggestions for Authors

  1. While the article itself is fine, a total lack of theoretical investigations of the structure using ab initio methods is slightly disappointing. It may be rather helpful in studying the structure and defects, preventing the manuscript from providing a more significant impact on the field.

We agree that ab initio modeling in most cases give additional independent information about structural inhomogeneities in single crystal materials. At the same time, I want to draw your attention that a significant part of the article is devoted to the modeling of defect formation at Ni+ ions stopping in YIG using individual interatomic potentials for all ion/atom collisions. We agree that ab initio molecular dynamics method is could be useful for modeling of radiation defects clusters  formation, however, it is not possible to compare such results with the methods carried out in our research high resolution X-ray diffraction and Mossbauer spectroscopy.

  1. Neon ion implantation of YIG is nothing ground-breaking new in the field, and therefore a more precise investigation of what happens exactly would be nice, but maybe out of the scope this time.

We agree that we agree that Ne-implanted YIG is not a fundamentally new experiment, however, in this article for the first time, the relationship between structural changes, magnetic microstructure and domain structure is established

  1. While the change in magnetic properties is interesting, it provides minor new insights (but they are there).

In our opinion, the article has a methodical and generalizing character as a comprehensive study of the magnetic and crystalline microstructure of magnetic single crystals with radiation defects.

Some minor comments to the article:

-) Mössbauer is misspelled

We took into account your recommendations and correct this incomprehensibility.

-) "Described defects doesn’t significantly", change to don't

The appropriate corrections have been made

-) the iron ion configuration (3d4sx) is given rather imprecise

I agree with you, but the determining of Fe-O chemical bond covalence degree using isomer shifts calibration is successfully used

11 October 2022                                                                               Date of article correction

Round 2

Reviewer 1 Report

This version has been improved. Publication is thus recommended.